# An intronic transposon insertion associates with a trans-species color polymorphism in Midas cichlid fishes

Claudius F. Kratochwil [1,3]✉, Andreas F. Kautt [1,4], Alexander Nater [1], Andreas Härer [1,5], Yipeng Liang [1,6], Frederico Henning [2] & Axel Meyer [1]✉

Polymorphisms have fascinated biologists for a long time, but their genetic underpinnings often remain elusive. Here, we aim to uncover the genetic basis of the gold/dark polymorphism that is eponymous of Midas cichlid fish (*Amphilophus* spp.) adaptive radiations in Nicaraguan crater lakes. While most Midas cichlids are of the melanic "dark morph", about 10% of individuals lose their melanic pigmentation during their ontogeny and transition into a conspicuous "gold morph". Using a new haplotype-resolved long-read assembly we discover an 8.2 kb, transposon-derived inverted repeat in an intron of an undescribed gene, which we term *goldentouch* in reference to the Greek myth of King Midas. The gene *goldentouch* is differentially expressed between morphs, presumably due to structural implications of inverted repeats in both DNA and/or RNA (cruciform and hairpin formation). The near-perfect association of the insertion with the phenotype across independent populations suggests that it likely underlies this trans-specific, stable polymorphism.

[1] Zoology and Evolutionary Biology, Department of Biology, University of Konstanz, Universitätsstrasse 10, 78457 Konstanz, Germany. [2] Department of Genetics, Institute of Biology, Federal University of Rio de Janeiro (UFRJ), Rio de Janeiro, Brazil. [3] Present address: Institute of Biotechnology, HiLIFE, University of Helsinki, Helsinki, Finland. [4] Present address: Department of Organismic and Evolutionary Biology, Harvard University, Cambridge, MA 02138, USA. [5] Present address: Division of Biological Sciences, Section of Ecology, Behavior & Evolution, University of California San Diego, La Jolla, CA, USA. [6] Present address: Department of Biology, University of Virginia, Charlottesville, VA 22903, USA. ✉email: Claudius.Kratochwil@helsinki.fi; Axel.Meyer@uni-konstanz.de

Ever since Edmund Brisco Ford defined polymorphisms as the co-occurrence of two or more distinct genetic forms of a species[1], it has been debated how such discontinuous variation is caused and maintained[2]. The presence of "gold" and "dark" individuals (Fig. 1a) in the Midas cichlid species complex (*Amphilophus* spp.) is a particularly striking example of a stable color polymorphism that has been studied for almost half a century[3,4]. The two-color morphs coexist in at least five out of seven Nicaraguan crater lake (CL) species assemblages, as well as in their source populations inhabiting the Great Lakes (GLs) Managua and Nicaragua. The gold morph is always uncommon, ranging from less than 1% to roughly 20% in different populations[5]. The gold/dark polymorphism is a Mendelian trait with the gold morph being the dominant form[5]. Up until young adulthood, "genetically" homo- or heterozygous gold (*GG* or *Gd*; with *G* being the "gold" and *d* the "dark" allele at the "gold locus") and dark (*dd*) individuals are both dark and visually indistinguishable. Between 3 months and 4 years of age or more[6], genetically "gold" individuals transition into the gold phenotype by losing the black pigmentation in their skin (usually within a few weeks) and become uniformly orange, red, yellow, or later sometimes even white[7]. During this process, melanophores, the melanin-bearing pigment cells, progressively undergo cell death[6].

Interestingly, this transition happens at a younger age in homozygous gold individuals than in heterozygotes[5]. Previous work using pedigree-based and cross-population genome-wide association mapping located the *gold* locus to chromosome 11 of a recently published chromosome-scale Midas cichlid genome assembly[5]. However, convincing candidates for the causally underlying mutation(s) or genes had been lacking so far.

Here we identify a novel transposon insertion that most likely constitutes the genetic basis of the gold/dark polymorphism as it almost perfectly associates with and predicts the two-color morphs. Moreover, we show that this large intronic insertion is likely to have a regulatory effect on the undescribed gene *gold-entouch*, presumably through its very large, conspicuous secondary structure.

## Results and discussion

**Conflicting results suggest a missing variant.** In order to narrow down candidates for the causal genetic variant, we performed genome-wide association mapping separately in individual lake populations (previously, association mapping was only performed across the whole species flock[5]). Interestingly, despite clear association peaks in the crater lakes (Fig. 1a, b), the exact position

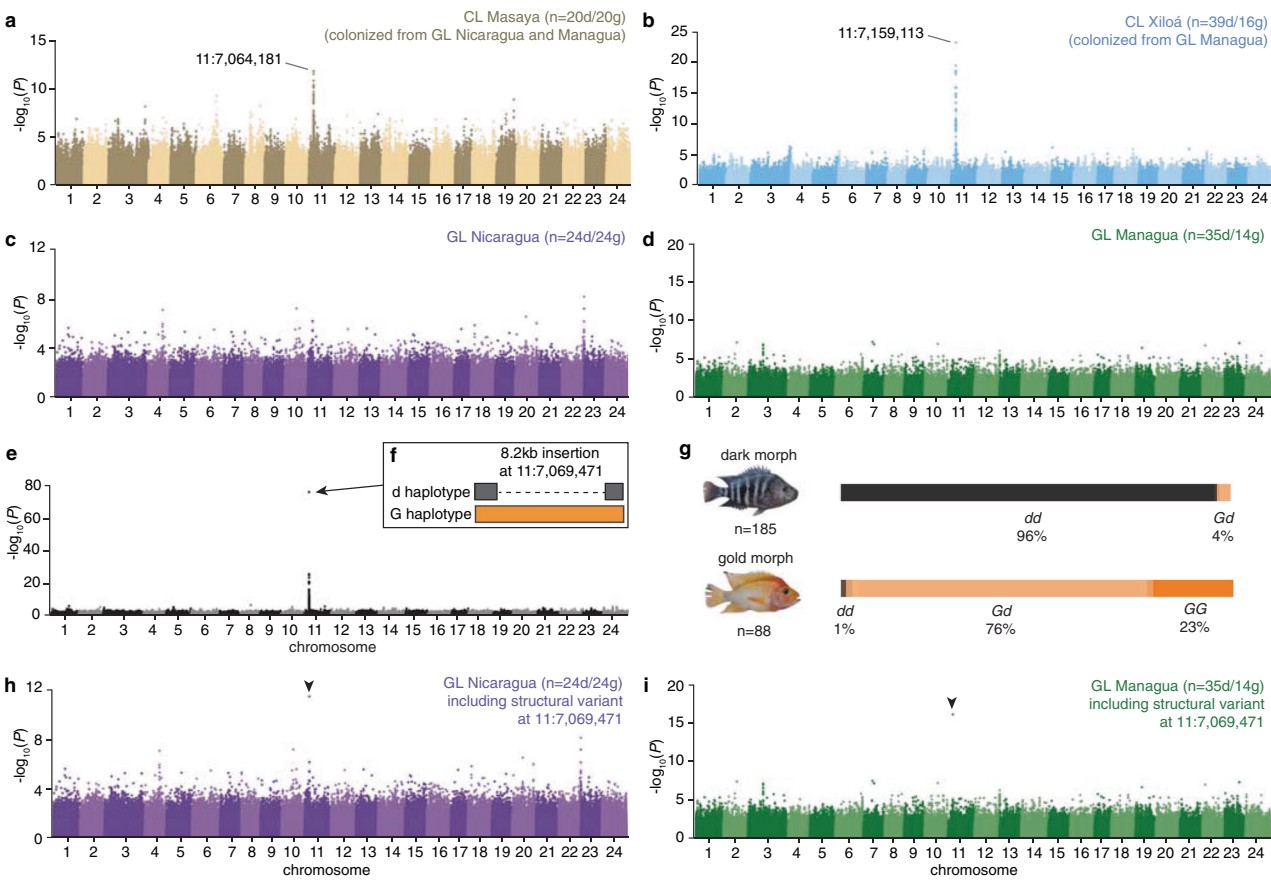

**Fig. 1 Genotype-phenotype mapping of the gold/dark polymorphism. a–d** Genome-wide association mapping for the gold/dark polymorphism in Midas cichlids using previous genotype data[5] reveals a prominent peak on chromosome 11 in CLs Masaya (**a**) and Xiloá (**b**) but not in the source populations from GL Nicaragua (**c**) and GL Managua (**d**). **e, f** The inclusion (**e**) of a newly discovered structural variant (**f**) in cross-population genome-wide association mapping (**e**) provides strong support that this variant constitutes the genetic basis of the phenotype (LRT test, $P = 2.23 \times 10^{-76}$). **g** Compatible with the dominant Mendelian inheritance pattern, individuals with one or two G alleles are gold, individuals with two d alleles are dark. Only 4% of dark individuals had the *Gd* genotype ($n = 8$), and likely represent phenotypically dark but yet untransformed genetically gold individuals (note that *Gd* individuals transform much later than *GG*s[5]). Among the golden individuals, we only found a single phenocopy that was gold but lacked the G allele (out of $n = 88$). Genotype distribution by morph and species/population is shown in Supplementary Fig. 1. **h, i** The analysis of the newly discovered structural variant (**f**) also reveals a much clearer association of this specific variant (arrowhead) with the phenotype in the two source populations from GL Nicaragua (**h**) GL Managua (**i**).

of the top-associated variants differed by ca. 95 kb between CL Masaya [11:7,064,181] and CL Xiloá [11:7,159,113]. Even more surprisingly, both great lake populations completely lacked strongly supported marker-trait associations (Fig. 1c, d). A possible explanation for both observations is that we solely identified linked variants and did not genotype the causal variant itself. This could be, for example, due to a structural variant (SV) that is either missing completely from the reference genome or is hard to identify due to limitations of the used methods (i.e., reads cannot be mapped or not reliably genotyped using short-read sequencing). In fact, SVs often remain uncovered, although they are a substantial source of population genetic variation and can have direct functional implications[8]. Owing to their relatively old population ages and large population sizes, linkage blocks are rather small in the Midas cichlid great lake populations. In these situations, it is expected that almost no association to neighboring markers will be detected, even to those that are in close proximity. Moreover, this could also explain the shifted association peaks in the crater lakes, as the most-highly associated linked variants can differ depending on the demographic history of the populations.

**Strong association of a newly identified structural variant**. To uncover structural variants that were possibly absent from the current reference genome (derived from a dark morph individual), we generated a de novo assembly of a lab-raised heterozygous golden individual of an $F_4$ inbred line using accurate long reads (PacBio HiFi). The assembly generated 817 primary scaffolds ($N_{50}$ of 2.47 Mb), with a total size corresponding to 95% of the Midas cichlid reference genome[4]. Both haplotypes at the gold locus were recovered, determined based on known variants at flanking sites that differed between the parents of the cross. The assemblies of the G and d haplotypes included the candidate interval, as well as hundreds of kilobases of flanking sequences (total size of the alignment of both haplotypes was 1.3 Mb). This new long-read alignment led to the discovery of an 8.2 kb insertion in the G haplotype at position 11:7,069,471 of the reference genome (Fig. 1f)—located in close proximity to the most-highly associated SNPs from our previous genome-wide association analysis (5.7 kb downstream; 11:7,063,765) and positioned between the peaks found in CLs Masaya (5.3 kb downstream; 11:7,064,181) and Xiloá (89.6 kb upstream; 11:7,159,113).

To test whether this newly identified structural variant is more strongly associated across all populations than the previously identified SNP/short indel markers, we added its genotype information to our previous genotype call set (see methods). When we repeated the genome-wide association mapping (Fig. 1e) we now obtained substantially stronger support for this variant (LRT test, $P = 2.23 \times 10^{-76}$; in comparison: the previously most-highly associated SNP using the same data except the one variant had statistical support of $P = 7.96 \times 10^{-27}$). Moreover, we also found strong support for this variant in the source lake populations from GL Nicaragua and GL Managua (Fig. 1h, i), where no strongly associated markers were found in the initial analysis (Fig. 1c, d). The overall association is near perfect (Fig. 1g, Supplementary Fig. 1), with only very few conflicting results explained by the characteristics of the focal trait. There are eight dark individuals with one copy of the G allele that are likely not yet transformed (4.3%; 8 of 185 individuals; Fig. 1b, Supplementary Fig. 1). Notably, all of these are heterozygotes, which is consistent with previous data showing that the onset of color change from dark to gold during ontogeny is dosage-dependent, with heterozygous fish transitioning much later in life (hence untransformed adult individuals would more likely carry the Gd genotype)[5]. Only a single genetically homozygous dark (dd) phenotypically golden individual was detected (0.6%; only 1

in 88 individuals, Fig. 1g, Supplementary Fig. 1). This represents very likely a genetic or non-genetic phenocopy.

***goldentouch* is differentially expressed between morphs**. The 8.2 kb insertion is located within an intron of *goldentouch*, a previously undescribed gene (Fig. 2a). Based on these new results and identification of the likely causal variant we hypothesized that the transposon insertion might affect expression of *goldentouch* (and/or possibly other genes in genomic proximity). To specifically screen for these genes as well as to identify downstream effector genes, we performed RNA-seq on scale tissue from adult gold ($n = 9$) and dark individuals ($n = 9$). Fish scales are covered by epidermal and dermal tissue that contain pigment cells including the dark melanophores that undergo cell death during the transition from dark to gold coloration[6]. In total, we found 247 differentially expressed genes (Fig. 2b Supplementary Data 1; $P_{adj} < 0.05$; total number of coding genes: 22,495). To screen for *cis*-regulatory variation at the gold locus, we specifically checked for differentially expressed genes within a 4 Mb window (variant ± 2 Mb; to also include long-range *cis*-regulated genes[9]) around the insertion site. Only one of the 96 genes in this region showed differential expression (Fig. 2c, d): the gene *goldentouch* that harbors the transposon insertion. A previous candidate gene underlying the gold/dark polymorphism, the gene *stk* that is directly 5′ of the locus did not show support for differential expression ($P = 0.99$). To provide further confirmation for the differential expression of *goldentouch*, we performed quantitative PCR (qPCR) in homozygous gold and dark individuals ($n = 6$ for each genotype). These results confirm a 2.4 times lower expression of this gene in the gold morph (Fig. 2e; Welch Two-Sample t-test [two-sided]; $P = 0.018$; $t = -3.04$, $df = 7.05$, 95% CIs, $-0.037$, $-0.004$).

***goldentouch* expression does not change during ontogeny**. As the morphological color change in Midas cichlids has an ontogenetic progression it is important and relevant to ask what genes change their expression during ontogeny. This has been investigated and many of the same pigmentation genes, as found here, have been previously identified[7]. We reanalyzed this data set using only genotypically golden individuals (before transition $n = 5$, transitional $n = 6$, after transition $n = 6$) using the Midas cichlid reference genome (the initial analysis[7] used two independent approaches; a de novo assembly and an assembly against the Nile tilapia transcriptome) to test two hypotheses. First, we tested if there are additional differentially expressed genes in any of the three pairwise comparisons within or close to the gold interval (2 Mb upstream and downstream). Although we found the same pigmentation genes (Supplementary Fig. 2a) that had been identified previously with this data set[7], we did not find any gene in or close to the gold interval (all genes in 4 Mb interval $P > 0.05$). Second, we wanted to specifically test if *goldentouch* expression changes during the process of color change (Fig. 2g). In contrast to melanophore marker genes (ANOVA, $P = 0.0003$, $P = 0.018$, $P = 0.0001$ for *mreg*, *tyrp1b*, and *pmela*; Fig. 2f) we did not find any significant expression changes for *goldentouch* throughout our ontogenetic sampling (ANOVA, $P = 0.37$; Fig. 2f). This lack of change was robust (ANOVA, $P = 0.31$; Supplementary Fig. 2b) even after accounting for genotype and only considering heterozygous Gd individuals (using markers in neighboring genes; sample size $n = 5$, $n = 4$, $n = 3$ for before, during and after transition). Furthermore, we found no significant correlation between melanin-related genes and *goldentouch* expression (linear regression, $P = 0.48$, $P = 0.19$, $P = 0.28$ for *mreg*, *tyrp1b*, and *pmela*; Supplementary Fig. 2c-h). This suggests that *goldentouch* is unlikely to be expressed in

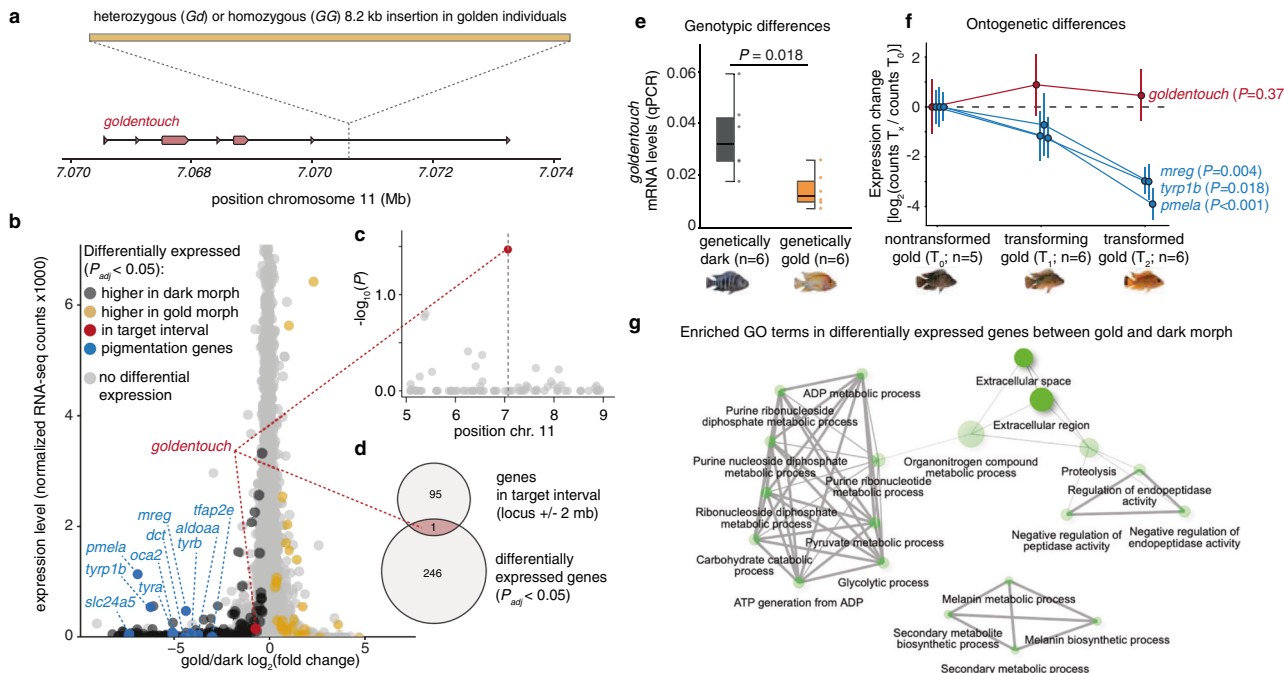

**Fig. 2 The genetic basis of the gold/dark polymorphism. a** The structural variant that likely underlies the gold/dark polymorphism is an 8.2 kb transposon insertion in the last intron of an heretoforth undescribed gene that we here coin *goldentouch*. **b–d** RNA-seq on scales of adult fish reveals 247 differentially expressed genes between gold and dark individuals ($P_{adj} < 0.05$; Benjamini–Hochberg adjusted Wald test *P*-value), including several pigmentation genes (**b**). The gene *goldentouch* is the only differentially expressed gene found within a 4 Mb interval around the insertion (**c, d**). **e** Quantitative PCR confirms the decreased expression of *goldentouch* in golden individuals (center line, median; box limits, upper and lower quartiles; whiskers, 1.5× interquartile range; points, data points; Welch's two-sample *t*-test, two-sided). **f** Relative change of gene expression during ontogeny in golden individuals. Expression of *goldentouch* (red) remains relatively stable, compared to melanophore marker genes (blue) that decrease in expression due to cell death of the melanophores in which they are expressed (data are presented as mean values ± SD; statistical test: one-way ANOVA). **g** Gene ontology (GO) term enrichment analysis of genes differentially expressed between gold and dark individuals.

melanophores themselves but in the surrounding skin tissue as the expression does not decrease with declining melanophore numbers. This data also provide insights into the mechanism as it might suggest that there is a constitutive, genotype-dependent expression of *goldentouch* that non-cell autonomously acts on melanophore survival.

**RNA-seq reveals downstream effectors of *goldentouch*.** To further explore the potential mechanisms behind the effect of *goldentouch* on melanophore (dark pigment cells) survival, we also examined potential downstream effects on genes outside the gold interval. Consistent with the observation that color transition involves the loss of cell types during the ontogeny in genetically gold individuals (e.g., melanophores), most differentially expressed genes (89%; 220 genes) were more highly expressed in the dark morph. Only 27 genes (11%) had higher expression in gold individuals (Fig. 2b). Indeed, many of the genes with lower expression in gold individuals are well-known melanophore-specific genes and key players in melanin synthesis (Fig. 2b), which is consistent with previous findings[7] and provides internal validation for our results. A gene ontology (GO) term analysis of the differentially expressed genes (Fig. 2g; Supplementary Table 1 and Supplementary Data 1) reveals an over-representation of genes associated with metabolism and extracellular space suggesting that non-cell-autonomous effects might indirectly lead to melanophore cell death. While the involvement of metabolic and proteolytic processes is more difficult to explain (it might indicate changes in skin homeostasis that indirectly affect melanophores[10]), interactions between cells and between cells and extracellular matrix are important for

pigment cell migration and survival[11–13]. Yet, the results generally hint at a complicated molecular mechanism underlying the morphological color change that might be rather due to a cumulative effect of non-cell-autonomous factors and their interaction with maturation and aging processes.

**Phylogenetic and structural analysis of *goldentouch*.** As the *goldentouch* gene had not been described before, we compared its coding sequence to the non-redundant NCBI nucleotide collection (nr/nt) using the tblastx algorithm to investigate whether (i) it is indeed a coding gene and (ii) whether it is also present in other species. We could identify orthologs of *goldentouch* in more than 20 teleost species, all of which belong to the Order Percomorpha (*sensu* ref. [14]). Thus, we suggest that the gene is a Percomorpha-specific gene that originated ~120 million years ago[14] (Fig. 3a). Owing to the lack of similarity to any other known genes it is possible that its origin was de novo. Next, we predicted the structure of *goldentouch* (Fig. 3b) and screened for more distant protein homologies and known domains. Our analysis gives weak support for a hydrolase/transport protein domain (confidence that the match between your sequence and this template is a homology: 55.0 on a scale from 0 to 100; Proportion of protein residues equivalenced to identical template residues in the generated alignment: 53%). However, the exact function of the protein within the cells of the dermis surrounding the melanophores remains unclear. Lastly, we analyzed the expression pattern of this gene across several tissues. Its expression was almost absent from most internal organs and highest in scales, supporting a specific function in the outer skin layers. (Fig. 3c).

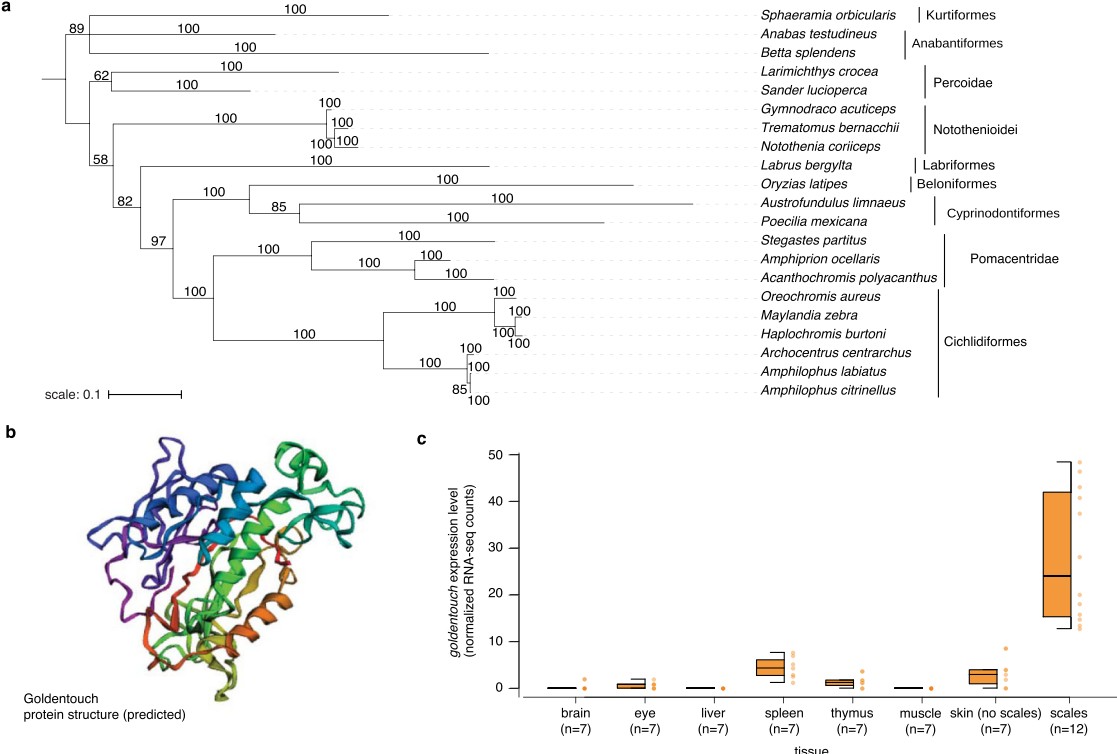

**Fig. 3 Phylogenetic, structural, and expression analyses of *goldentouch*. a** Bayesian phylogenetic reconstruction of *goldentouch*, which was only found in the order Percomorpha and could, for example, not be detected in zebrafish, *Danio rerio*, a non-Percomorpha. **b** Three-dimensional structure prediction of *goldentouch*. **c** Expression of *goldentouch* is mostly restricted to scales, with some expression in the underlying skin and in spleen tissue (center line, median; box limits, upper and lower quartiles; whiskers, 1.5× interquartile range; points, outliers).

**The insertion is an inverted repeat with a 4.1 kb stem**. A further important question is how the transposon insertion might downregulate *goldentouch* expression. Comparing the sequence against the non-redundant NCBI nucleotide collection (nr/nt) using blastx revealed that the insertion is transposon-derived and shows high similarity to a DNA or class II transposon, PiggyBac Transposable Element Derived 4. Interestingly, the transposon is duplicated in the sequence and thereby appears as a symmetrical structure in a plot of the blastx result: a dotplot revealed a secondary structure with two hairpins with a 4.1 kb stem (Fig. 4a). Such structures that form cruciforms (Fig. 4b) have been shown to affect DNA supercoiling and nucleosome positioning[15]. The stalling of Polymerase II elongation would lead to mRNA without poly-adenylation likely resulting in unstable, quickly degraded mRNA, explaining the lower expression in golden individuals. While RNA without poly-adenylation would have not at all been picked up by our RNA-seq experiment, the formation of a large hairpin on the pre-mRNA would be expected to decrease translation. This is supported by our qPCR results, suggesting that it affects mRNA stability (and or expression itself), either through lack of the polyA tail or through premature termination due to the hairpin structure of the repeat. It remains to be investigated whether this SV also affects other features of this locus including somatic as well as meiotic mutation rates as well as recombination rate as had been suggested as a consequence for long inverted repeat (LIR) sequences[16]. DNA properties (in this case Z-DNA formation) have been, for example, recently shown to greatly affect the dynamics and to facilitate adaptive repeated evolution in sticklebacks[17].

Additionally, independent of the effect on DNA conformation, the presence of the transposon within an actively transcribed region might affect *goldentouch* expression. Antisense transcription of transposons is also effective in blocking sense transcription[18]. Moreover, many mechanisms exist (including heterochromatin spreading and by stalling Polymerase II elongation) by which host cells actively silence TE expression, as well as that of the genes into which they integrated[19]. Furthermore, given the established and drastic implications of such an inverted-repeat TE, it is likely that a multitude of mechanisms, such as those that we proposed above, play a role. Functional validation, including knockout of the transposable element—which is challenging given its size—would help to establish a causal link that could confirm our findings.

**The developmental mechanism of the gold/dark polymorphism**. Although the causal link needs further confirmation and the exact molecular mechanisms remains elusive up to now, our results seem to support the notion that secondary DNA structures and/or transposable elements can be an important driver of (adaptive) evolution. Other cases in the last decades have illustrated how transposons can affect gene regulation and phenotypic evolution: Transposon insertions have been, for example, associated with insecticide resistance in *Drosophila*[20], industrial melanism in the peppered moth[21], and sex-linked color polymorphisms in birds[22]. What makes the Midas cichlid case so interesting and seemingly unique until now is that it seems to be such a surprisingly complex and intricate molecular process that leads to the loss of pigment cells in adult cichlids. After all, many other mutations could contribute to the process of turning Midas cichlids gold, including melanophore markers, such as *tyrosinase*, *oca2*, or members of the Agouti/Mc1r pathway that have been shown to abolish melanin production in cichlids[23–25]. Ultimately, the Midas cichlid gold locus will help address longstanding questions on the proximate and ultimate causes of stable (color) polymorphisms. Future research will show why this insertion and the mechanism by which it acts to trigger pigment loss at a

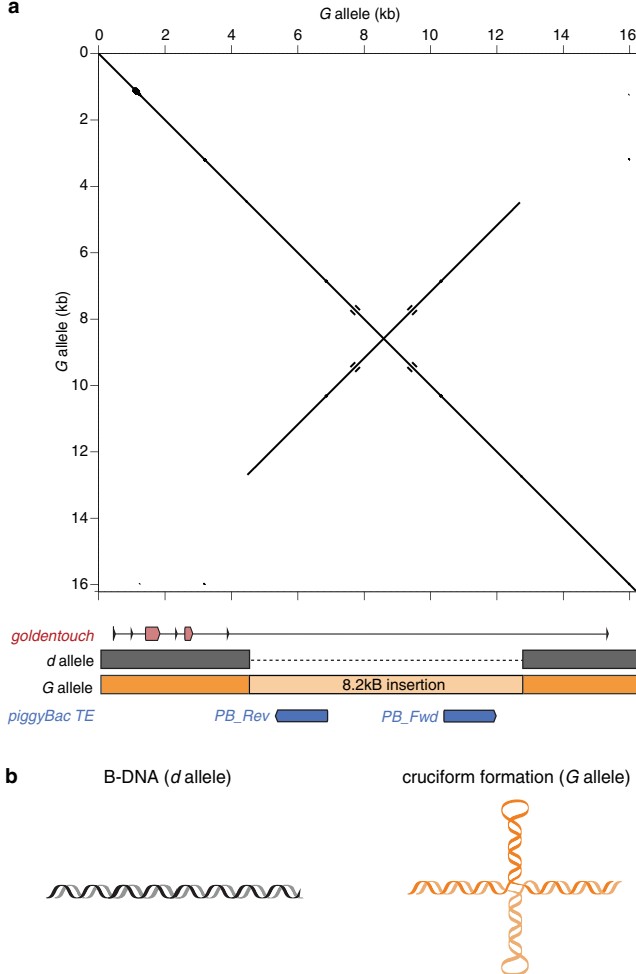

**Fig. 4 Structural properties of the structural variant in gold Midas cichlids. a** A dotplot of the golden allele at the genomic position of the *goldentouch* gene. The structural variant we identified based on a haplotype-resolved PacBio HiFi assembly of a heterozygous golden individual is an 8.2 kb transposon-derived inverted repeat with identical copies of a piggyBac transposable element (TE). **b** The two alleles likely differ in their secondary structure as inverted repeats are known to form cruciform structures. The part of the two copies that form the stem of the DNA cruciform have sequence similarity and differ only at two positions (two single-base-pair indels).

specific ontogenetic phase has been maintained by selection in so many Midas cichlid populations. Our preliminary data on the ecological relevance of this, to the human eye at least, much more conspicuous Gold morph in this color polymorphism might suggest that the gold morphs are maintained at low frequencies in all populations because predators do not seem to prefer them[26].

In summary, we identified what might constitute the genetic basis of the trans-species gold/dark polymorphism that is eponymous of Midas cichlids: a transposon-derived structural variant that likely causes de-regulation of a previously undescribed gene *goldentouch* via the formation of a huge cruciform secondary structure. We anticipate that the knowledge of the likely causal mutation will not only greatly contribute to deciphering the molecular mechanism of this fascinating case of morphological color change but also lead to exciting avenues of research on the interaction between genomic structural features, assortative mating[27,28], selection regimes[29], and ecological settings[26] that lead to the stable persistence of intraspecific polymorphisms[3,4].

## Methods

**PacBio assembly of a heterozygous mapping panel individual.** To screen for structural variants, a phenotypically gold *Gd* heterozygous male from an $F_4$ inbred line (cross described in ref. [4]) was used for accurate long-read sequencing (PacBio Hifi) and de novo assembly. Extraction and PacBio HiFi sequencing were performed at Praxis Genomics (Atlanta, GA). DNA was extracted from peripheral blood using the SP Blood & Cell DNA Isolation kit (Bionano) and was sheared to between 15 kb and 23 kb using g-TUBES (Covaris). PacBio HiFi sequencing libraries were prepared using the SMRTbell™ Express Template Prep Kit 2.0, purified using the SMRTbell Enzyme Cleanup Kit and were size-selected using a SageELF (SAGE Science). The appropriate fractions for sequencing runs were identified on the 5200 Fragment Analyzer (Agilent) and pooled to make up the sequencing libraries, with a final cleanup and concentration step using AMPure PB beads (Pacific Biosciences). Prior to sequencing, all libraries were checked for concentration using Qubit™ 1X dsDNA HS Assay Kit (Thermo Fisher) and final size distribution was confirmed on the 5200 Fragment Analyzer (Agilent). SMRTbell libraries were bound to the sequencing polymerase enzyme using the Sequel II Binding Kit 2.0, per manufacturer's instructions with the modification that the Sequencing Primer v2 was annealed to the template instead of the standard primer. Prior to sequencing, the unbound polymerase enzyme was removed using a modified AMPure PB bead method. Following 2 h of pre-extension, shotgun genomic DNA sequence data were collected on the Pacific Biosciences Sequel II system using HiFi sequencing protocols and Sequencing kit V2 for 30 h to allow ample time for multiple pass sequencing and yield high-quality circular consensus sequencing (HiFi) results.

Raw reads were processed to generate accurate long reads using ccs v.5.0.0 (https://github.com/PacificBiosciences/pbbioconda). A total of 2.97 M reads were obtained, with a subread N50 of 16.6 Kb and a total molecular yield of 49.33 Gb. Of the total reads, 1.28 M passed filters for generating HiFi reads (minimum of three passes), resulting in ~21 Gb of HiFi reads (~22x reference genome coverage). HiFi reads were assembled using Improved Phased Assembler (IPA) v.1.3.0 (https://github.com/PacificBiosciences/pbipa), including phasing and polishing steps. The gold locus was detected in an 8 Mb scaffold that showed high collinearity with the reference and a segment of 1,334,965 bp representing the alternate haplotype (from the final.a_ctg.fasta output from IPA).

**Genome-wide association mapping.** We performed single variant phenotype-genotype association mapping using the linear mixed model approach implemented in GEMMA v0.98.1[30] using the -lmm 4 option. Relatedness was accounted for by including a centered kinship matrix as a random effect. We used the same genotype file as previously used in ref. [4]. and adopted the same filtering strategy of excluding variants with minor allele frequency less than 0.05 and more than 20% missing data per analysis (e.g., lake-wise) (accession numbers: genome, JAC-BYM000000000; bam files, PRJEB38173). We did not apply a Hardy-Weinberg-Equilibrium filter, to avoid any chance of over-filtering potentially relevant or even causal variants. To perform genotype–phenotype mapping with the new variant we genotyped the structural variant and added it to the previous genotype call set. We decided on this strategy as the previous reference genome had superior quality and was already annotated. In addition, this approach allowed for a better comparison of the analyses (as the datasets only differ in genotypes for a single, added variant position: the transposon insertion). To genotype the insertion, we calculate the ratio of soft-clipped reads (indicative of the insertion) and total reads that overlapped with the insertion position 11:7069471 using a custom python script[31] and pysam (https://github.com/pysam-developers/pysam). We excluded reads that were flagged as unmapped or duplicates and those overlapping the breakpoint position by fewer than 5 base pairs in both directions since these would be unreliable for our calling purposes (soft-clipped reads vs. matching read). Reads indicative of harboring the G allele were then defined as those containing soft-clipped positions (as indicated in CIGAR string) near the insertion start breakpoint (±5bp to account for random matches). Individuals with ratios of split to total reads below 0.1 (or an absolute split-read number ≤ 2) where scored as homozygous *dd*, individuals with ratios above 0.9 (or an absolute total_read—split-read number ≤ 2) as homozygous *GG*. All other individuals were scored as heterozygotes (*Gd*). Genotyping was verified by manual inspection of the bam files at this position. In all illustrations we used the $-\log_{10}$ transformed *P* values derived from LRT test.

**RNA sample collection, isolation, and library preparation.** Scales were collected from 18 lab-raised (to reduce environmental effects) adult Midas cichlid individuals, derived from stocks of wild-caught fish from CLs Masaya, Apoyeque, and Xiloá in Nicaragua. Gold and normal stocks from the three different populations were kept separately. Other tissues were taken from heterozygous individuals from the gold mapping panel[4]. For scales for each individual, two scales were removed from the center of the trunk and kept in RNAlater (Invitrogen) at −20 °C for long-term storage. The unpigmented anterior part of the scales was discarded before RNA extraction. RNAlater was removed prior to homogenization. All tissues were homogenized in 200 μl Promega-supplied lysing buffer in 2 ml Lysing Matrix A tubes (MP Biomedicals) using FastPrep-24 Classic Instrument (MP Biomedicals), RNA extraction and on-column DNase treatment were performed with the

ReliaPrep miRNA Cell and Tissue Miniprep System (Promega). Following extraction, RNA was quantified using the Qubit RNA HS Assay Kit (Invitrogen) with a Qubit Fluorometer (Life Technologies). The RNA integrity number (RIN) was checked using an RNA 6000 Pico Kit (Agilent) on a 2100 Bioanalyzer System (Agilent). RNA-seq libraries were prepared using the TruSeq Stranded mRNA Library Prep Kit (Illumina) according to the manufacturer's protocol. Briefly, 1 μg RNA was put into mRNA selection by poly-T oligo attached magnetic beads followed by fragmentation (94 °C for 6 min). The cleaved mRNA fragments were reverse transcribed into first-strand cDNA using GoScript Reverse Transcriptase (Promega) and random hexamer primers (Illumina). We used Illumina-supplied consumables to synthesize second-strand cDNA followed by adenylating 3′ ends. Barcoded adapters from TruSeq RNA CD Index Plate (Illumina) were ligated to the ends of the double-strand cDNA. The final libraries were amplified using 15 PCR cycles and quantified and quality-assessed using an Agilent DNA 12000 Kit on a 2100 Bioanalyzer (Agilent). Indexed DNA libraries were normalized. Libraries were sequenced on a HiSeq X Ten platform (BGI Genomics, Beijing).

**Mapping, data processing and RNA-seq data analysis**. Adapters were trimmed from raw reads using trimmomatic v0.38[32]. Processed raw reads were aligned to the *Amphilophus citrinellus* genome (accession number JACBYM000000000) and annotation[5] using the STAR RNA-seq aligner v2.6.1d[33]. Expression counts were calculated with RSEM v.1.3.3[34] using the above-mentioned annotation. The data were subsequently analyzed in R using the DESEQ2 v1.22.1 pipeline[35] in the R environment. As individuals were from different lakes, we corrected by the lake in the DEseq2 design formula. The reported *P*-values are Benjamin–Hochberg FDR corrected, as implemented in DEseq2. GO Term analysis was performed using the ShinyGO v.0.66 pipeline[36] using the Nile tilapia (*O. niloticus*) genome as a reference and standard settings (0.05 FDR *P*-value threshold; all available gene sets) showing the top 20 pathways. To re-genotype individuals used in the previous RNA-seq experiment[7], we used markers in neighboring genes (as the individuals were from the mapping panel for which we had genotypes based on the new assembly). Based on the allele ratios we could unequivocally genotype the individuals as *dd*, *Gd*, and *GG* individuals.

**Quantitative PCR**. Total RNA was reverse transcribed with the GoScript Reverse Transcription System (Promega, Madison, Wisconsin). Quantitative Real-Time PCR (qPCR) was performed to determine relative expression levels for *goldentouch* in scale tissue. After an initial denaturation step (95 °C for 2 min), qPCR reactions were run for 40 cycles (95 °C for 15 s, 60 °C for 1 min) (CFX96 Real-Time System; Bio-Rad Laboratories, Hercules, California) using specifically designed primers (goldentouch_fwd: 5′-AGC TGC TCA AAC GAC ACT GA; goldentouch_rev: 5′-ACG CCA ACC ACA ATC ACA ATG). The total volume for each reaction was 20 μl consisting of 2 μl cDNA (5 ng/μl), 0.5 μl forward primer (10 μM), 0.5 μl reverse primer (10 μM), 10 μl GoTaq qPCR Master Mix, 2× (GoTaq qPCR Master Mix; Promega, Madison, Wisconsin) and 7 μl Nuclease-free H$_2$O. All qPCR reactions were followed by a melt curve analysis to test for amplification specificity. Expression levels were quantified using mean threshold cycle (Ct) values based on three and two technical replicates for target genes and housekeeping genes, respectively. Relative gene expression of the target gene was calculated using the geometric mean of two housekeeping genes (ldh2_fwd: 5′-TTG GAG GTT TTG AGG AAA AGG; ldh2_rev: 5′-: CAG GAA CAA GGT GAC TGT GGT; imp2_fwd: 5′-GCC TGG AGC ATG TTG ACC; imp2_rev: 5′-CGA AGT GAC GGA TCT TAC GG).

**Phylogenetic analysis, other genomic analyses, and protein prediction**. For generating a phylogenetic tree of *goldentouch* we used the NGPhylogeny pipeline[37] with MAFFT v7.407[38] for alignment, BMGE 1.12[39] for sequence curation and Mr. Bayes v.3.2.6[40] for Bayesian phylogenic inference. Alignment and dotplot were generated using MAFFT[38]. To search for known domains in *goldentouch* we used the Protein Homology/analogY Recognition Engine V 2.0 (Phyre[2]) using the predicted amino acid sequence of *A. citrinellus* with the intensive modelling mode. To compute the three-dimensional structure, we used trRosetta using the same sequence[41]. The software is based on de novo folding, guided by deep learning restraints. The confidence of the predicted model was however low (estimated TM-score = 0.236).

**Ethics oversight**. Euthanasia of animals and animal husbandry were approved by the German authorities (permit numbers T-16/13, Regierungspräsidium Freiburg, Abteilung 3, Referat 35, Veterinärwesen & Lebensmittelüberwachung, Germany).

**Reporting summary**. Further information on research design is available in the Nature Research Reporting Summary linked to this article.

## Data availability
The RNA-seq data generated in this study have been deposited in the Sequence Read Archive database under accession codes PRJNA635556 and PRJNA766232. The Pac-Bio raw reads data used in this study are available in the Sequence Read Archive database under accession code PRJNA694028. All other relevant raw and intermediate data

(PacBio assembly, bam files of the target region, sample lists, gene and protein sequences of *goldentouch*) used in this study are available in the Dryad database [https://doi.org/10.5061/dryad.zs7h44j9p][42].

## Code availability
Code is available at Zenodo (https://doi.org/10.5281/zenodo.5528090)[43] and Github (https://github.com/akautt/Midas_gold_dark_GWAS.git)[31].

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

## Acknowledgements

This work was supported by the European Research Council (ERC advanced grant GenAdapt 293700) and grants by the Deutsche Forschungsgemeinschaft (DFG) to A.M. (219669982), by a European Molecular Biology Organization to A.F.K. (ALTF 47-2018), by a Swiss National Science Foundation fellowship to A.N. (P300PA_177852), by grants from Instituto Serrapilheira (Serra-529) and CAPES (PrInt-PGGen) to F.H., and the Baden-Württemberg Foundation, the Deutsche Forschungsgemeinschaft (290977748, 423396155) and startup funding of the Institute of Biotechnology, University of Helsinki to C.F.K. We gratefully thank Jean-Nicolas Volff and Ville Paavilainen for fruitful discussion on the transposon and protein computation part, respectively.

## Author contributions

Long-read sequencing and assembly: F.H.; genomic analysis: A.F.K., C.K., A.N., and F.H.; funding acquisition: A.M., F.H., and C.F.K.; transcriptome generation and analysis: C.F.K., A.F.K., Y.L., and A.H., conceptualization: C.F.K., F.H., A.F.K., and A.M.; Quantitative PCRs: A.H., Phylogenetic, structural analyses: C.F.K.; manuscript draft and figures: C.F.K., F.H., and A.F.K. All authors contributed to, improved, and approved the final manuscript.

## Funding

## Competing interests

The authors declare no competing interests.
