## [Peer Review File · Nature Communications]

An intronic transposon insertion associates with a trans-species color polymorphism in Midas cichlid fishesReviewers' Comments:

Reviewer #1:

Remarks to the Author:

This paper uncovers the genetic basis of a polymorphic colour trait in African Cichlids (*Amphilophus* spp.), by 1) de novo assembly of a genome from an individual whose phenotype makes it certain that the causal locus should be included in the assembly, and 2) GWAS, carried out in different populations (ancestral and derived) of the fish.

The logic of the progression from first GWAS -> new genome -> second GWAS is nice, and it's pleasing that it worked out so convincingly. At this point, from the main text, I was a bit unclear on the details of the second GWAS – was the structural variant added to the previous mapped data as a single locus whose genotype was known, or were the reads remapped to the new reference assembly? (I think it must be the former but I'm not sure this is clear enough in the main text).

The paper follows up the GWAS with functional investigations of the locus, which is a transposable element insertion into an intron of a previously unannotated gene. I also found the main text slightly lacking in detail here - how was the gene annotated de novo in this study? Then transcriptomic analysis confirmed that the gene was differentially expressed between black and gold morphs, and its expression was highest in scale tissue. I don't feel qualified to comment on the suitability of the methods for the functional part of the paper.

Overall, the work is convincing and the biology of the system is really interesting. I think that the identification of a TE insertion that likely affects expression of a gene in a way that impacts fitness in the wild is super-interesting, but is not unprecedented. Indeed, at some point, I wonder if the authors should make reference to other examples of transposable elements effecting gene expression in an adaptive context- Daborn et al (2002) is the first one that springs to mind, Kim et al (2019) is another, but I am an author on that paper so I don't insist on its inclusion, although it does explicitly deal with balancing selection, which is relevant to this system.

I agree with the final paragraph of the main text, and I'm looking forward to reading future published work on this polymorphism, particularly its evolutionary significance. Some specific comments below:

Lines 96-97:

1) Could the authors please present the name / a description of the statistical test that the GWAS p-values are derived from, not just its significance, here in the main text?

2) I'm not clear whether the two P-values are directly comparable. It would help if the authors were more explicit about the exact differences between the datasets that the two GWAS were performed on here. Is it the same assembly and SNP data, and the only difference is that the insertion genotype has been included as one extra locus? Or were the reads remapped to the new assembly for the second test? I'm wondering if (hypothetical) correction for multiple testing might be different for the two datasets.

3) A minor point: could the y-axis scales for Supplementary Figure 1a-d be the same as those for e-h, for each corresponding pair of plots?

Line 109:

1) "PiggyBac Transposable Element Derived 4" – This came out of blue, somewhat.

2) How was the "unknown gene" first annotated, please – could this be explained briefly in the main

text at, or before, this point? I am specifically interested in the step(s) that presumably occurred before the BLASTX search of the NCBI data.

Lines 144-146: Supplementary Figure 3d is pretty convincing regarding higher expression in the scales – you could possibly make it clearer here in the main text that the difference is real?

Line 164: I don't think that the authors mean "encipher" here. "decipher"?

None of the original code used to carry out the analyses is available, as far as I can see? It would be better if it was, including to reviewers.

References

Daborn, P.J., Yen, J.L., Bogwitz, M.R., Le Goff, G., Feil, E., Jeffers, S., Tijet, N., Perry, T., Heckel, D., Batterham, P. and Feyereisen, R., 2002. A single P450 allele associated with insecticide resistance in *Drosophila*. *Science*, 297(5590), pp.2253-2256.

Kim KW, Jackson BC, Zhang H, Toews DP, Taylor SA, Greig EI, Lovette IJ, Liu MM, Davison A, Griffith SC, Zeng K. Genetics and evidence for balancing selection of a sex-linked colour polymorphism in a songbird. *Nature communications*. 2019 Apr 23;10(1):1-1.

Reviewer #2:

Remarks to the Author:

How intraspecific polymorphism arises in a population and is maintained in the population is crucial for understanding the initial step of speciation accompanying phenotypic evolution.

Kratochwil et al. addressed this issue by performing genome-wide association studies to identify the crucial gene for the Gold/Black body color polymorphism, one of the conspicuous traits associated with speciation in Midas cichlid fishes in Nicaraguan crater lakes. Their genetic data based on the wild populations and a laboratory-maintained strain narrowed down the causal genetic regions into a ~95-kb scale DNA fragment located on the 11th chromosome in the previous reference genome sequence. Additional de novo genome assembly based on accurate long reads (PacBio HiFi) identified an 8.2 kb transposon insertion in an intron of an unknown gene located in the ~95-kb region, which they named goldentouch, in the Golden(G) allele. Their comparative mRNA-seq analysis between the adult scale tissues in Golden(G) and Dark(D) adults revealed that goldentouch is the only gene significantly differentially expressed between the G and D alleles among the ~95 kb candidate region. Besides, they also found that goldentouch was a taxon-specific gene (Percomorpha-specific). Taken together, they concluded that goldentouch is a putative causative gene for Gold/Black body color polymorphism in the Midas cichlid fishes.

Overall, I found that their population genetic analyses are compelling. Their conclusion that a taxon-specific (Percomorpha-specific) unknown gene is involved in polymorphism formation is intriguing from evolutionary biology's viewpoint. Besides, their finding might lead to an understanding of the novel molecular mechanism of pigmentation in vertebrates. Still, I think it would be hasty to conclude that the goldentouch gene is responsible for the body color polymorphism in Midas cichlid fishes solely based on transcriptome data in the adult tissues.

Major concerns:

1. Loss-of-function analysis and, if possible, gain-of-function analysis of the unknown gene they named goldentouch in the Midas cichlid are necessary to conclude that an intronic insertion into the goldentouch gene affected its expression to generate the body color polymorphism. A cis-regulatory

modification sometimes results in modified expression in several surrounding genes (e.g., Pardo-Diaz & Jiggins, 2014, *Evol Dev*). Besides, comparative transcriptome analysis only in a single adult stage can lead to the misestimation of the causal gene. I believe it deserves more efforts on characterizing the function of the unknown gene in the Midas cichlid using the genome editing (CRISPR/Cas9) available in laboratory-maintained cichlid fishes (Kratochwil et al. 2018).

2. Related to the function of the goldentouch gene, I think the authors' sampling for mRNA-seq analysis after coloration is too late for comparative analysis because the causal gene could be expressed in the melanophores only before they undergo cell death. As explained in the manuscript, melanophores are eliminated by cell death during the transition from dark to gold coloration in the GG or GD individuals. Therefore, I suggest that the authors should collect scale tissues at the onset or at least in the middle of color transition when the subset of melanophores to be eliminated are still alive and reevaluate the transcriptome data. This experiment would be realized by periodic tissue sampling (e.g., regular sampling in 10-day intervals) during the color development and retrograde decision of which sample to be analyzed for mRNA-seq using the laboratory-maintained strain.

3. The most disappointing point is the absence of a clear justified evolutionary model mediated by intronic transposon insertion in the manuscript. The authors should discuss this point because the intronic transposon insertion is in the title of this paper. Recent advances in sequence technologies have enabled us to estimate functions of non-coding DNA sequences in the genome readily (e.g., ATAC-seq, enhancer RNA sequencing, and so forth). The authors should at least present sequencing data to precisely evaluate the function of intronic insertion in the goldentouch gene to estimate how an intronic insertion has led to phenotypic evolution (e.g., driven by the acquisition of a novel enhancer or an alternative exon). Ultimately, it would be ideal for the authors to perform transgenic enhancer assays focusing on the inserted element to show in which cell type the inserted piggyBac element drives the expression of goldentouch gene.

Minor issues:

1. False Discovery Rate (q-value) should be used to estimate differentially expressed genes in transcriptome analysis associated with Figure 2b because comparative analysis using RNA-seq datasets should consider the effect of multiple comparisons.

2. I recommend the authors include a diagram depicting the predicted domains in the Goldentouch protein, including transmembrane domains, in figure 2a to help readers readily understand its structure.

Reviewer #3:

Remarks to the Author:

The authors identified a structural variant in the Midas cichlid species complex by de novo genome assembly of gold individuals, when investigating a discrepancy in genome-wide associations for the gold/dark polymorphism. The gold haplotype contains a transposon-driven insertion, and by comparing differentially expressed genes to genes near the transposon, the authors identified a gene they name goldentouch. The authors highlight pigmentation genes among the differentially expressed genes as validation of the RNA-seq analysis and propose a model where reduction of goldentouch expression directly or indirectly triggers melanophore apoptosis. Finally, the authors characterised goldentouch by its phylogenetic distribution, protein structure, and expression in different tissues, finding it to be specific to Percomorpha and expression highest in scales.

The paper well-reasoned, suitable evidenced, and convincing. The findings described are novel, interesting, and of significance to the field. In terms of flow of the paper, Extended Data Figure 1 could be provided in the main body as it demonstrates the motivation for the work, and the paper would be easier to follow without the current need to alternate between the main and extended data.

Extended Data Figure 3 would not be out of place in the main text as well. Another small writing note, the name goldentouch could be introduced earlier in the text since it is named early in the figures; this would make it easier to follow where the gene is mentioned in the text.

Regarding the conclusions and claims, the use of the word "underlies" in the title is misleading. The gene and TE insertion identified are most likely responsible for the polymorphism. However, without functional validation the transposon being linked to a different, causal SNP cannot be ruled out. Therefore, a more appropriate phrase to use in the title would be "is associated with" or similar.

The authors suggest that highlighting differentially expressed genes with pigmentation roles is an exploration of the mechanism of melanophore death in gold individuals, but this provides little insight into the mechanism per se. The reduction in melanophore-associated genes reflects the ultimate effect of melanophore death in gold individuals and, as the authors note, provides internal validation. It may be better to avoid the use of the word mechanism here. Conversely, the full list of differentially expressed genes could provide insights into the mechanism downstream of goldentouch expression reduction, but this full list is not provided. The full list of differentially expressed genes should be published, as well as the genome assembly and the raw reads. Accession numbers are missing for example, this article should not be published without them.

The number of samples is not reported for the qPCR for expression levels of the gene in different tissues. The distribution of data points for this could be shown alongside the box plots in the Extended Data Figure 3 (as was done for Figure 2, which was good to see.)

The methodology is sound and meets the expected standards in the field. The software tools are appropriate choices, widely used. The 1Mb window for narrowing down the differentially expressed genes can be reasonably expected to cover any genes the transposon insertion may be affecting. However there were a couple of points that could be clarified. In the RNA-seq on scale tissues, the genotypes were not specified, only the phenotypes. Could this mean that some dark individuals are heterozygous and would have later turned gold? It would be good to include an explanation about why the genotypes were not provided or not known for these samples. Also the method of designating heterozygotes seems very broad (anything between 0.1 and 0.9 for the proportion of split reads to matched reads) – could the reasoning be explained further?

This paper adds evidence that transposable elements are important for rewiring gene expression. There are not that many cases in adaptive evolution where this can be seen so clearly, so this paper provides an interesting example.

Point-to-point reponse

Reviewer 1

1.1 *This paper uncovers the genetic basis of a polymorphic colour trait in African Cichlids (Amphilophus spp.), by 1) de novo assembly of a genome from an individual whose phenotype makes it certain that the causal locus should be included in the assembly, and 2) GWAS, carried out in different populations (ancestral and derived) of the fish. The logic of the progression from first GWAS -> new genome -> second GWAS is nice, and it's pleasing that it worked out so convincingly.*

Our response: Thank you for the concise summary of our work.

1.2 *At this point, from the main text, I was a bit unclear on the details of the second GWAS – was the structural variant added to the previous mapped data as a single locus whose genotype was known, or were the reads remapped to the new reference assembly? (I think it must be the former but I'm not sure this is clear enough in the main text).*

Our response: This should have been emphasized more clearly. But you are right — we added the structural variant to the previous mapped data. Reasons for this are twofold. Firstly, the assembly published in Kautt et al. 2020 is more contiguous since it was scaffolded with additional data (BioNano optical maps and Hi-C). Lastly, we wanted to be able to compare the new and old results (i.e. with and without the structural variant — see also 1.8). In our revised version we now provide more details in the main text as well as and include our reasoning for choosing this approach in the methods section.

1.3 *The paper follows up the GWAS with functional investigations of the locus, which is a transposable element insertion into an intron of a previously unannotated gene. I also found the main text slightly lacking in detail here - how was the gene annotated de novo in this study?*

Our response: The gene was found in the in-house generated annotation of our new chromosome-level assembly (Kautt et al., 2020), but is, until now, undescribed (including ENSEMBL or NCBI databases)

1.4 *Then transcriptomic analysis confirmed that the gene was differentially expressed between black and gold morphs, and its expression was highest in*

scale tissue. I don't feel qualified to comment on the suitability of the methods for the functional part of the paper.

Our response: Thank you for the transparency. Reviewers 2 and 3 both commented on the RNA-seq part and they deemed the methods to be suitable.

- 1.5 *Overall, the work is convincing and the biology of the system is really interesting. I think that the identification of a TE insertion that likely affects expression of a gene in a way that impacts fitness in the wild is super-interesting, but is not unprecedented. Indeed, at some point, I wonder if the authors should make reference to other examples of transposable elements effecting gene expression in an adaptive context- Daborn et al (2002) is the first one that springs to mind, Kim et al (2019) is another, but I am an author on that paper so I don't insist on its inclusion, although it does explicitly deal with balancing selection, which is relevant to this system. Then transcriptomic analysis confirmed that the gene was differentially expressed between black and gold morphs, and its expression was highest in scale tissue. I don't feel qualified to comment on the suitability of the methods for the functional part of the paper.*

Our response: Thank you for the positive comment and also the helpful remarks. We agree and have now included a more comprehensive discussion, including the suggested references

- 1.6 *I agree with the final paragraph of the main text, and I'm looking forward to reading future published work on this polymorphism, particularly its evolutionary significance.*

Our response: Thank you for this kind comment. We are also excited for these future avenues of research.

- 1.7 *(1) Could the authors please present the name / a description of the statistical test that the GWAS p-values are derived from, not just its significance, here in the main text?*

Our response: Apologies for the lack of detail. The analysis has been described more thoroughly in the revised version. Briefly, we used the LRT test, which is preferred as it makes fewer approximations (<https://github.com/genetics-statistics/GEMMA/issues/86>). We also uploaded the full output to dryad (<https://datadryad.org/stash/share/Gct0O3XJJs5D1IHpAeE8F18MuKICGW1MCNNJYaj6N0>).

1.8 (2) *I'm not clear whether the two P-values are directly comparable. It would help if the authors were more explicit about the exact differences between the datasets that the two GWAS were performed on here. Is it the same assembly and SNP data, and the only difference is that the insertion genotype has been included as one extra locus? Or were the reads remapped to the new assembly for the second test? I'm wondering if (hypothetical) correction for multiple testing might be different for the two datasets.*

Our response: GWA was performed independently for the transposon genotype calls and simply added to the previous GWA. P-values are comparable since they are derived from variant-by-variant analysis using the same method (i.e. p-values are uncorrected and controlled for multiple testing only to define the significance threshold. Thus, the results are identical except for that one new data point (which is negligible for multiple testing concerns). This has been mentioned more clearly now.

1.9 (3) *A minor point: could the y-axis scales for Supplementary Figure 1a-d be the same as those for e-h, for each corresponding pair of plots?*

Our response: Thank you for this suggestion. We now adjusted the y-axis and also moved the panels to the first main figure (as suggested by Reviewer 3).

1.10 *Line 109: "PiggyBac Transposable Element Derived 4" – This came out of blue, somewhat.*

Our response: Agreed. This comment prompted us to analyze this insertion in more detail, which generated crucial insights. We provide a bit more context in our revised version and devote now a whole paragraph to this.

1.11 *How was the "unknown gene" first annotated, please – could this be explained briefly in the main text at, or before, this point? I am specifically interested in the step(s) that presumably occurred before the BLASTX search of the NCBI data.*

Our response: Our latest in-house generated annotation, which we published last year together with the chromosome-level reference genome (Kautt et al. 2020), but we did not have strong evidence that it is indeed a coding gene (which our analyses here now provides strong evidence for).

1.12 *Lines 144-146: Supplementary Figure 3d is pretty convincing regarding higher expression in the scales – you could possibly make it clearer here in the main text that the difference is real?*

Our response: Thank you very much for this suggestion. We emphasize this more clearly in the main text now.

1.13 Line 164: I don't think that the authors mean "encipher" here. "Decipher"?

Our response: Correct, we meant decipher.

1.14 None of the original code used to carry out the analyses is available, as far as I can see? It would be better if it was, including to reviewers.

Our response: Raw and intermediate data as well as code have been uploaded now and accession numbers are provided (PRJNA694028, PRJNA635556, <https://doi.org/10.5061/dryad.zs7h44j9p> (not yet live, reviewer URL is here: <https://datadryad.org/stash/share/Gct0O3XJJs5D1IHpAeE8F18MuKICGW1MCNNJYaj6N0>) and https://github.com/akautt/Midas_gold_dark_GWAS.git).

Reviewer 2

2.1 *How intraspecific polymorphism arises in a population and is maintained in the population is crucial for understanding the initial step of speciation accompanying phenotypic evolution. Kratochwil et al. addressed this issue by performing genome-wide association studies to identify the crucial gene for the Gold/Black body color polymorphism, one of the conspicuous traits associated with speciation in Midas cichlid fishes in Nicaraguan crater lakes. Their genetic data based on the wild populations and a laboratory-maintained strain narrowed down the causal genetic regions into a ~95-kb scale DNA fragment located on the 11th chromosome in the previous reference genome sequence. Additional de novo genome assembly based on accurate long reads (PacBio HiFi) identified an 8.2 kb transposon insertion in an intron of an unknown gene located in the ~95-kb region, which they named goldentouch, in the Golden(G) allele. Their comparative mRNA-seq analysis between the adult scale tissues in Golden(G) and Dark(D) adults revealed that goldentouch is the only gene significantly differentially expressed between the G and D alleles among the ~95 kb candidate region. Besides, they also found that goldentouch was a taxon-specific gene (Percomorpha-specific). Taken together, they concluded that goldentouch is a putative causative gene for Gold/Black body color polymorphism in the Midas cichlid fishes.*

Our response: Thank you for this excellent summary of our work.

2.2 Overall, I found that their population genetic analyses are compelling. Their conclusion that a taxon-specific (*Percomorpha*-specific) unknown gene is involved in polymorphism formation is intriguing from evolutionary biology's viewpoint. Besides, their finding might lead to an understanding of the novel molecular mechanism of pigmentation in vertebrates. Still, I think it would be hasty to conclude that the *goldentouch* gene is responsible for the body color polymorphism in *Midas* cichlid fishes solely based on transcriptome data in the adult tissues.

Our response: Thank you for the kind words about our population genetic analyses. Regarding the role of *goldentouch* we did not want to give the impression that it is unequivocal that this is indeed the causal gene. This is why we were careful in our phrasing and wrote “likely genetic basis” — but we agree that our title was too strongly phrased, which we have therefore changed accordingly to “..is associated with...”. Still, we would like to emphasize that we think that deregulation of this gene is the most parsimonious explanation, because of the lack of other differentially expressed genes and because of the fact that this very large insertion with a newly discovered secondary structure will have very likely an effect on gene expression. Our new findings, that we include in this revised version of the manuscript now provides a mechanistic explanation for this deregulation (formation of a cruciform/hairpin structure — see below) that further supports our hypothesis.

2.3 Major concerns:

1. Loss-of-function analysis and, if possible, gain-of-function analysis of the unknown gene they named *goldentouch* in the *Midas* cichlid are necessary to conclude that an intronic insertion into the *goldentouch* gene affected its expression to generate the body color polymorphism. A cis-regulatory modification sometimes results in modified expression in several surrounding genes (e.g., Pardo-Diaz & Jiggins, 2014, *Evol Dev*). Besides, comparative transcriptome analysis only in a single adult stage can lead to the misestimation of the causal gene. I believe it deserves more efforts on characterizing the function of the unknown gene in the *Midas* cichlid using the genome editing (CRISPR/Cas9) available in laboratory-maintained cichlid fishes (Kratochwil et al. 2018).

Our response: Surely CRISPR-Cas9 would be the gold standard to provide fully convincing evidence that the effects of the transposon insertion in *goldentouch* is indeed causal. A knockout of an insertion of that size (over 8kb) is, however, nearly impossible in a non-model organism such as *Midas* cichlids. A knockout of the gene might be feasible, but these experiments have never been done in this species, are still very time consuming because of the late transition (often

>1 year), and might prove to be unsuccessful because of early lethality or because the full knockout does not mimic the low-expression phenotype. Lastly, first generation (mosaic) knockouts will likely not even be sufficient to generate the phenotype. The required stable knockout lines would be very difficult to obtain in Midas cichlids (2-3 years).

Nevertheless, as also Reviewer 1 points out, our evidence for a causal role of the transposon is strong and comparable to other well known studies such as the peppered-moth (Van't Hoff 2016), which is also the main message (and hence title) of our work as we agree that the support for the associated indel is higher than for the gene that might be affected by it. Based on the location of the insertion and the expression difference, we however do think that *goldentouch* is the most likely gene to be affected by the insertion (that is almost perfectly associated with the phenotype). Probably the best argument — that we expanded on in our revised version — is that we are likely not seeing a “traditional” cis-regulatory effect caused by regulatory elements. Expression is reduced by the insertion (golden individuals have the insertion and low expression levels of *goldentouch*). Therefore the expression difference is likely not caused by enhancers that transposed to the locus together with the PiggyBac TE. We speculate that the reduced expression is instead most likely caused by the insertion itself, that causes a 4.1kB cruciform structure that will affect transcription and translation (see 2.5). We agree that loss or gain-of-function evidence would be desirable, but given the difficulty and quantity of experiments that would have to be conducted we believe acquiring those data is beyond the scope of this study. Instead, we hope that by having toned down our conclusion, we are now striking the balance of providing strong evidence for an exciting mechanistic hypothesis, without overstating our results.

- 2.4 *2. Related to the function of the goldentouch gene, I think the authors' sampling for mRNA-seq analysis after coloration is too late for comparative analysis because the causal gene could be expressed in the melanophores only before they undergo cell death. As explained in the manuscript, melanophores are eliminated by cell death during the transition from dark to gold coloration in the GG or GD individuals. Therefore, I suggest that the authors should collect scale tissues at the onset or at least in the middle of color transition when the subset of melanophores to be eliminated are still alive and reevaluate the transcriptome data. This experiment would be realized by periodic tissue sampling (e.g., regular sampling in 10-day intervals) during the color development and retrograde decision of which sample to be analyzed for mRNA-seq using the laboratory-maintained strain.*

Our response: Until now we don't really know the exact molecular mechanisms and at what stage (and in which cell type) expression differences cause the

phenotype. Regular sampling throughout the transition is a fine approach but (beside the substantial time investment of another 1–2 years) very challenging as it is very difficult to sample at the right time before the transition (which starts somewhere between 6 and 18 months in our fish). What we have done in the past (Henning et al. 2013) is to conduct RNA-seq at three stages, before (T_0) during (T_1) and after (T_2) the transition. These analyses did not discover any differentially expressed genes within the genomic interval around the gold locus. To answer the question if there is expression change of *goldentouch* during the process of color change, for this revised manuscript we reanalyzed the transcriptome data set using our recent reference genome and annotation (previous analyses were done both, *de novo* and based on a Nile tilapia reference transcriptome). These results do not provide evidence for variation between stages ($P=0.37$, ANOVA) suggesting that the color change is not induced by ontogenetic changes in *goldentouch* expression. The lack of correlation between melanophore marker genes and *goldentouch* furthermore provides evidence that the gene is itself not expressed in melanophores. How exactly the gene expression difference results in the phenotypic differences remains to be shown (but this was not the focus of this work), but using gene ontology (GO) term analysis of the differentially expressed genes suggest that melanophores might die due to non-cell-autonomous effects from the extracellular surroundings.

2.53. *The most disappointing point is the absence of a clear justified evolutionary model mediated by intronic transposon insertion in the manuscript. The authors should discuss this point because the intronic transposon insertion is in the title of this paper. Recent advances in sequence technologies have enabled us to estimate functions of non-coding DNA sequences in the genome readily (e.g., ATAC-seq, enhancer RNA sequencing, and so forth). The authors should at least present sequencing data to precisely evaluate the function of intronic insertion in the goldentouch gene to estimate how an intronic insertion has led to phenotypic evolution (e.g., driven by the acquisition of a novel enhancer or an alternative exon). Ultimately, it would be ideal for the authors to perform transgenic enhancer assays focusing on the inserted element to show in which cell type the inserted piggyBac element drives the expression of goldentouch gene.*

Our response: As this was intended as a short format we wanted to be concise and not expand on the mechanism. In our revised version we provide a very straight-forward mechanism for the downregulation of *goldentouch*. As the insertion is an almost perfect inverted repeat, it is prone to form a cruciform/hairpin structure with a 4.1kB stem. This will greatly affect the gene both on the transcriptional and translational level. Based on the known effects

of hairpins on transcription we therefore strongly believe that this is a sufficiently justified model to explain the reduced expression of *goldentouch*.

Although transposons have been often suggested to harbour regulatory elements we do not think that this is likely in this case as the insertion reduces gene expression in gold individuals. We also don't have any evidence for an alternate exon. The suggested silencing mechanism above seems the most likely explanation. Mapping of ATAC-seq reads to a transposable element would also have a lot of caveats because reads could not be mapped specifically. The generation of a stable transgenic GFP line with such a large insert (we would be dealing with a >15kB plasmid) that might still actively transpose is an extremely challenging endeavour — even in zebrafish — that, if feasible at all, could take several years in Midas cichlids.

Our main focus here was to present the likely causal mutation of the gold-dark polymorphism. The full characterization of the molecular mechanism is a super exciting research avenue which we are already planning, but will require several years to unravel. Instead, we believe that this finding of a SV that explains such an evolutionarily relevant trait represents a valuable contribution to the literature and deserves publication as such. In our revised manuscript we state this more clearly now, but suggest the mechanism (cruciform/hairpin formation) by which the gene (*goldentouch*) is likely affected. Nonetheless, we agree that more experimental work is needed to confirm this, which we also state in the manuscript.

2.6 *Minor issues:*

1. False Discovery Rate (q-value) should be used to estimate differentially expressed genes in transcriptome analysis associated with Figure 2b because comparative analysis using RNA-seq datasets should consider the effect of multiple comparisons.

Our response: The p-value presented is using the Benjamin-Hochberg FDR as implemented in DEseq2. We used the abbreviation P_{adj} instead of FDR or q-value as this corresponds to the output column of DEseq2.

2.7 2. I recommend the authors include a diagram depicting the predicted domains in the Goldentouch protein, including transmembrane domains, in figure 2a to help readers readily understand its structure.

Our response: Thank you. We have added the information to the main figure. In our revised version we included also a new reconstruction. We do not highlight the transmembrane domains anymore, as it was suggested to us by an expert that these are very difficult to predict with high accuracy (especially if there is

such a lack of homology as in our case). The size of the protein makes it more likely that it is a cytosolic protein. We corrected this in the main text.

Reviewer 3

3.1 *The authors identified a structural variant in the Midas cichlid species complex by de novo genome assembly of gold individuals, when investigating a discrepancy in genome-wide associations for the gold/dark polymorphism. The gold haplotype contains a transposon-driven insertion, and by comparing differentially expressed genes to genes near the transposon, the authors identified a gene they name goldentouch. The authors highlight pigmentation genes among the differentially expressed genes as validation of the RNA-seq analysis and propose a model where reduction of goldentouch expression directly or indirectly triggers melanophore apoptosis. Finally, the authors characterised goldentouch by its phylogenetic distribution, protein structure, and expression in different tissues, finding it to be specific to Percomorpha and expression highest in scales.*

Our response: That is a very nice summary of our work. Thank you very much.

3.2 *The paper well-reasoned, suitable evidenced, and convincing. The findings described are novel, interesting, and of significance to the field. In terms of flow of the paper, Extended Data Figure 1 could be provided in the main body as it demonstrates the motivation for the work, and the paper would be easier to follow without the current need to alternate between the main and extended data. Extended Data Figure 3 would not be out of place in the main text as well. Another small writing note, the name goldentouch could be introduced earlier in the text since it is named early in the figures; this would make it easier to follow where the gene is mentioned in the text.*

Our response: Thank you for this helpful comment. We added the information to the main text and mention the gene name earlier.

3.3 *Regarding the conclusions and claims, the use of the word “underlies” in the title is misleading. The gene and TE insertion identified are most likely responsible for the polymorphism. However, without functional validation the transposon being linked to a different, causal SNP cannot be ruled out. Therefore, a more appropriate phrase to use in the title would be “is associated with” or similar.*

Our response: While we were careful in the main text, we agree that underlies is too strong as we merely show a correlation. In our revised version we have

changed the title to "...is associated with..." and also discuss the possibility that we are still missing genetic variants and that functional validation will be needed to confirm these results.

- 3.4 *The authors suggest that highlighting differentially expressed genes with pigmentation roles is an exploration of the mechanism of melanophore death in gold individuals, but this provides little insight into the mechanism per se. The reduction in melanophore-associated genes reflects the ultimate effect of melanophore death in gold individuals and, as the authors note, provides internal validation. It may be better to avoid the use of the word mechanism here. Conversely, the full list of differentially expressed genes could provide insights into the mechanism downstream of goldentouch expression reduction, but this full list is not provided.*

Our response: Yes, we still lack an exact mechanisms, which is the subject of future work and will surely take several years. Here, we provide evidence that the transposon insertion affects the expression of goldentouch. What remains unclear is how exactly the transposon insertion and goldentouch cause the melanophore loss. Although we find many additional differentially expressed genes (that we discuss now more comprehensively in the revised version) they do not give a clear indication of the mechanism (which however was not our focus here). At this point we can say that goldentouch is likely not expressed in melanophores itself, as it does only vary between genotypes but not throughout ontogeny. Enriched gene ontology terms show an overrepresentation of genes associated with metabolism and extracellular space, suggesting that non-cell autonomous effects might underlie the melanophore cell death. While the involvement of metabolic processes is more difficult to explain (but might indicate changes in skin homeostasis that indirectly affect melanophores), interactions between cells and between cells and extracellular matrix have been shown to be important for pigment cell migration and survival. A better understanding of the molecular function of *goldentouch* will hopefully help to get more insights into the mechanism that at this point is still very difficult to understand in its full complexity.

- 3.5 *The full list of differentially expressed genes should be published, as well as the genome assembly and the raw reads. Accession numbers are missing for example, this article should not be published without them.*

Our response: Raw and intermediate data as well as code have been uploaded now and accession numbers are provided (PRJNA694028, PRJNA635556, <https://doi.org/10.5061/dryad.zs7h44j9p> (not yet live, reviewer URL is here:

<https://datadryad.org/stash/share/Gct0O3XJJs5D1IHpAeE8F18MuKICGW1MCNNJYaj6N0>) and https://github.com/akautt/Midas_gold_dark_GWAS.git).

3.6 *The number of samples is not reported for the qPCR for expression levels of the gene in different tissues. The distribution of data points for this could be shown alongside the box plots in the Extended Data Figure 3 (as was done for Figure 2, which was good to see.)*

Our response: Apologies for this oversight. We now included the sample sizes for Extended Figure 3d (now Figure 3c) and show the distribution.

3.7 *The methodology is sound and meets the expected standards in the field. The software tools are appropriate choices, widely used. The 1Mb window for narrowing down the differentially expressed genes can be reasonably expected to cover any genes the transposon insertion may be affecting. However there were a couple of points that could be clarified. In the RNA-seq on scale tissues, the genotypes were not specified, only the phenotypes. Could this mean that some dark individuals are heterozygous and would have later turned gold? It would be good to include an explanation about why the genotypes were not provided or not known for these samples.*

Our response: In our revised version we expanded the analysis on the neighboring genes (now 4Mb, 2Mb upstream and downstream) and plotted the whole interval. Unfortunately, we do not have genotypes for these samples. Yet, we only sampled dark individuals from dark parents, so we can exclude any untransformed individuals.

3.8 *Also the method of designating heterozygotes seems very broad (anything between 0.1 and 0.9 for the proportion of split reads to matched reads) – could the reasoning be explained further?*

Our response: While the threshold between Gd and GG is not as relevant (as the phenotype is dominant), we wanted to be very conservative for the dd individuals. As a few softclipped reads are already indicative of a heterozygous individuals we set the ratio threshold very close to 0 at 0.1. We also implemented a threshold based on the absolute number of reads (≤ 2 softclipped reads overlapping the insertion breakpoint for dds) to account for samples with extraordinary high read depth for which a ratio-based approach would be less sensitive (e.g. if 4 out of 50 reads showed evidence for the inversion, that would be below the 0.1 ratio threshold, but a DD sample is unlikely to contain several (filtered) softclipped reads that span the insertion breakpoint by chance. The code is now provided on GitHub under

| https://github.com/akautt/Midas_gold_dark_GWAS.git. In addition, we visually inspected all BAM files at this position and confirmed the automatic scoring.

3.9 *This paper adds evidence that transposable elements are important for rewiring gene expression. There are not that many cases in adaptive evolution where this can be seen so clearly, so this paper provides an interesting example.*

| Our response: Thank you for this kind and positive assessment.

Reviewers' Comments:

Reviewer #1:

Remarks to the Author:

I am happy with the authors' responses to my original comments.

Reviewer #2:

Remarks to the Author:

My major concerns on the previous manuscript were not resolved because the authors provided no additional genetic data or developmental transcriptome/epigenome data between alleles. Without functional genetic validation and proper developmental transcriptome/epigenetic analysis, we geneticists often fail to indicate the causal gene for a specific phenotype, which can lead to a wrong conclusion.

1. Needs for functional genetic assays

The authors must at least show the phenotype of G0 individuals of CRISPR-Cas9-mediated knockout experiments (crispants) to indicate that goldentouch is responsible for pigmentation. Although the authors mentioned that goldentouch might be a lethal gene, we can readily avoid the embryonic lethality of lethal genes by inducing somatic mosaic knockout cell clones, including a single whole scale when we properly titrate the concentration of the CRISPR-Cas9 constructs for injection. The authors could have performed this experiment in the last six months before submitting this revised version of their manuscript, or at least should have provided the result indicating whether goldentouch is a lethal gene or not. Otherwise, the authors should have tested the function of the goldentouch ortholog gene by genome editing in one of the genetics model fishes, *Oryzias latipes*, whose generation time is about three months. Genetic functional validation is essential for publication in genetic studies.

2. Characterization of the effect of transposon insertion

The 4.1-kb-scale large palindrome structure of the inserted transposable element that the authors found in the revised manuscript is intriguing. Still, the authors provided no evidence supporting that the insertion at the goldentouch gene that regulates scale development is essential for phenotypic polymorphism. The resolution of genetic linkage analysis is still not enough to conclude that the transposon insertion at the goldentouch locus is essential for the polymorphic pigmentation phenotype. The authors' results cannot rule out other DNA sequence variations in the 95kb-region is essential for the color pattern polymorphism. The only data that the authors provided for the evidence for the function of the transposon insertion is mRNA-seq data for the adult scale tissue in the G and D alleles, which I think that cannot rule out the possibility that another developmentally expressed (transcribed and translated) gene located in the 95-kb region is differentially expressed between the alleles and is essential for the polymorphic phenotype. Without functional genetic validation of the goldentouch gene or epigenetic developmental information near the goldentouch intron, the model that the authors showed in Figure 4 sounds only handwaving because genome editing and NGS data are readily available in recent genetic studies.

An additional minor concern:

In my previous review comment, I recommended including the Goldentouch protein's domain structure in the figure (including the information about how long the domain is, how many known domains or transmembrane domains are included in the protein, and so forth). Such information is necessary for the readers to understand what kind of protein the goldentouch encodes. In addition, I noticed that the authors did not provide any amino acid sequence of the Goldentouch protein, nor are the sequence data for Goldentouch deposited to a public database. The authors should provide the amino acid sequence of Goldentouch so that the readers can readily extend their own research based on the sequence information of Goldentouch.

Reviewer #3:

Remarks to the Author:

The authors addressed all our concerns. I find the paper extremely interesting and recommend it for publication.

Point-to-point reponse

Reviewer 1

1.1 *I am happy with the authors' responses to my original comments.*

Our response: We thank reviewer 1 again for their constructive comments in round 1 and are pleased that our revisions were satisfactory.

Reviewer 2

2.1 *My major concerns on the previous manuscript were not resolved because the authors provided no additional genetic data or developmental transcriptome/epigenome data between alleles. Without functional genetic validation and proper developmental transcriptome/epigenetic analysis, we geneticists often fail to indicate the causal gene for a specific phenotype, which can lead to a wrong conclusion.*

Our response: Our main conclusion is that we find a near-perfect association (in 97% we find a match between phenotype and genotype; most of the non-matching individuals are likely explained by not having transitioned yet) and highly significant ($P=2.23 \times 10^{-76}$) of a transposon insertion with the phenotype across several independent populations. Based on this strong association, that is further supported by previously published linkage data and the fact that the association holds across several (10!) independent populations/species we suggest that this insertion likely underlies this trans-specific, stable polymorphism. We agree with the reviewer that we cannot be sure that we indeed found the causal mutation and gene but we provide strong evidence for our interpretation — further research will show, if we were right. Some of the most prominent examples of alleles that underlie adaptive traits are still at this stage, where despite all evidence we might deal with linked and not the ultimately causal or at least not solely responsible alleles. We carefully went through our maintext again and paid attention that our interpretations do not overstep the data.

In our revised version (R1) we did provide a developmental transcriptome analysis, which in our opinion gives some initial insights into the –seemingly complicated– developmental mechanisms.

2.2 *Needs for functional genetic assays. The authors must at least show the phenotype of G0 individuals of CRISPR-Cas9-mediated knockout experiments (crispants) to indicate that goldentouch is responsible for pigmentation. Although the authors mentioned that goldentouch might be a lethal gene, we can readily avoid the embryonic lethality of lethal genes by inducing somatic mosaic knockout cell clones, including a single whole scale when we properly titrate the concentration of the CRISPR-Cas9 constructs for injection. The authors could have performed this experiment in the last six months before submitting this revised version of their manuscript, or at least should have provided the result indicating whether goldentouch is a lethal gene or not. Otherwise, the authors should have tested the function of the goldentouch ortholog gene by genome editing in one of the genetics model fishes, *Oryzias latipes*, whose generation time is about three months. Genetic functional validation is essential for publication in genetic studies.*

Our response: The suggested experiments are indeed possible, but we deemed them not very likely to lead to a fast positive and/or readily interpretable result. We believe it crucial to first understand more about the molecular and developmental mechanisms to choose an appropriate genome editing strategy. As we are likely dealing with a non-cell-autonomous phenotype, mosaics will very likely not be sufficient. Knockouts in *Oryzias latipes* that we first would have to establish, are worth trying, but we seemed it unlikely that it will cause the same phenotype in a species that has a divergence time of approx. 129 million years (and we do not know of a comparable phenotype outside of the Midas cichlid system). In summary, we think that these experiments on a phenotype as complicated as the gold-dark-polymorphism cannot be rushed. Based on our experience, these experiments would almost certainly need at least four to five years to be rigorously done. Small-scale experiments that cannot be as well controlled might quickly lead to wrong conclusions. Since we believe that our claims are substantially supported, we deem it important to make our findings publicly available, so that the gene can be readily investigated in other species.

2.3 *Characterization of the effect of transposon insertion. The 4.1-kb-scale large palindrome structure of the inserted transposable element that the authors found in the revised manuscript is intriguing. Still, the authors provided no evidence supporting that the insertion at the goldentouch gene that regulates scale development is essential for phenotypic polymorphism. The resolution of genetic linkage analysis is still not enough to conclude that the transposon insertion at the goldentouch locus is essential for the polymorphic pigmentation phenotype. The authors' results cannot rule out other DNA sequence variations in the 95kb-region is essential for the color pattern polymorphism. The only data that the authors provided for the evidence for the function of the transposon insertion is*

mRNA-seq data for the adult scale tissue in the G and D alleles, which I think that cannot rule out the possibility that another developmentally expressed (transcribed and translated) gene located in the 95-kb region is differentially expressed between the alleles and is essential for the polymorphic phenotype. Without functional genetic validation of the goldentouch gene or epigenetic developmental information near the goldentouch intron, the model that the authors showed in Figure 4 sounds only handwaving because genome editing and NGS data are readily available in recent genetic studies.

Our response: Our main conclusion is that there is a near-perfect association of a transposon insertion with the phenotype across several independent populations. We have looked for almost ten years for sequence and transcriptome variation at this locus. Having examined four long-read assemblies and over 450 resequenced genomes using alignments as well as output from screens for structural variation we believe it is unlikely that we missed any/major variants in this interval. It is still possible that the insertion is not the causal one and that there is an equally well or even better associated variant in close proximity but we do not deem that to be likely. In any way, we describe the by far most-associated variant so far ($P=2.23 \times 10^{-76}$; 97% of the individuals match — the nonmatching ones are presumably explained by late-transitioning individuals) and provide all the necessary data to prove us wrong. The involvement of the goldentouch gene and the proposed mechanisms are indeed more speculative and we expanded on these during the revision, but we make it very clear that these are conjectures at this point; conjectures that should facilitate further analyses by providing falsifiable hypotheses.

Without wanting to compare our study to this one, one of the most famous examples of regulatory evolution and adaptive evolution in non-model organisms is the *pitx1* enhancer that causes pelvic fin reduction in sticklebacks. The QTL data was published in 2004 in Nature, the regulatory mechanism 2010 in Science and the underlying molecular mechanism that causes the enhancer deletion in 2019 in Science. Still today, the complete developmental mechanism is not fully understood. Yet, it was important to publish the milestones on the way as they crucially influenced the thinking and research of a whole field.

- 2.4 *An additional minor concern: In my previous review comment, I recommended including the Goldentouch protein's domain structure in the figure (including the information about how long the domain is, how many known domains or transmembrane domains are included in the protein, and so forth). Such information is necessary for the readers to understand what kind of protein the goldentouch encodes. In addition, I noticed that the authors did not provide any amino acid sequence of the Goldentouch protein, nor are the sequence data for Goldentouch deposited to a public database. The authors should provide the*

amino acid sequence of Goldentouch so that the readers can readily extend their own research based on the sequence information of Goldentouch.

Our response: In our previous point-to-point response to this comment we stated that we do not highlight the transmembrane domains anymore, as it was suggested to us by an expert that these are very difficult to predict with high accuracy (especially if there is such a lack of homology as in our case). The same is true for the hydrolyse domain that we mention in the text, but do not want to highlight in the figure as this would be misleading (because of the relatively poor support). Sequence data for goldentouch has been already provided in the first revision (the whole alignment of all goldentouch sequences was provided). We agree that it would be helpful for other researchers to also make the amino acid sequence of the Goldentouch protein available, which we now provide in this revised version of the manuscript.

Reviewer 3

3.1 The authors addressed all our concerns. I find the paper extremely interesting and recommend it for publication.

Our response: Thank you for the positive assessment. We are pleased we could address all of your concerns in our revised version.